# Improving Machine Learning Classification Accuracy for Breathing Abnormalities by Enhancing Dataset

**DOI:** 10.3390/s21206750

**Published:** 2021-10-12

**Authors:** Mubashir Rehman, Raza Ali Shah, Muhammad Bilal Khan, Syed Aziz Shah, Najah Abed AbuAli, Xiaodong Yang, Akram Alomainy, Muhmmad Ali Imran, Qammer H. Abbasi

**Affiliations:** 1Department of Electrical Engineering, HITEC University, Taxila 47080, Pakistan; 18-phd-ee-002@student.hitecuni.edu.pk (M.R.); raza.ali.shah@hitecuni.edu.pk (R.A.S.); 2Department of Electrical and Computer Engineering, COMSATS University Islamabad, Attock Campus, Attock 43600, Pakistan; engr_tanoli@ciit-attock.edu.pk; 3Research Centre for Intelligent Healthcare, Coventry University, Coventry CV1 5FB, UK; syed.shah@coventry.ac.uk; 4College of Information Technology, United Arab Emirates University (UAEU), Abu Dhabi 15551, United Arab Emirates; najah@uaeu.ac.ae; 5School of Electronic Engineering, Xidian University, Xi’an 710071, China; xdyang@xidian.edu.cn; 6School of Electronic Engineering and Computer Science, Queen Mary University of London, London E1 4NS, UK; a.alomainy@qmul.ac.uk; 7School of Engineering, University of Glasgow, Glasgow G12 8QQ, UK; Muhammad.Imran@glasgow.ac.uk; 8Artificial Intelligence Research Centre (AIRC), Ajman University, Ajman 20550, United Arab Emirates

**Keywords:** RF sensing, OFDM, CSI, SDR, USRP, COVID-19, breathing patterns

## Abstract

The recent severe acute respiratory syndrome coronavirus 2 (SARS-CoV-2), also known as coronavirus disease (COVID)-19, has appeared as a global pandemic with a high mortality rate. The main complication of COVID-19 is rapid respirational deterioration, which may cause life-threatening pneumonia conditions. Global healthcare systems are currently facing a scarcity of resources to assist critical patients simultaneously. Indeed, non-critical patients are mostly advised to self-isolate or quarantine themselves at home. However, there are limited healthcare services available during self-isolation at home. According to research, nearly 20–30% of COVID patients require hospitalization, while almost 5–12% of patients may require intensive care due to severe health conditions. This pandemic requires global healthcare systems that are intelligent, secure, and reliable. Tremendous efforts have been made already to develop non-contact sensing technologies for the diagnosis of COVID-19. The most significant early indication of COVID-19 is rapid and abnormal breathing. In this research work, RF-based technology is used to collect real-time breathing abnormalities data. Subsequently, based on this data, a large dataset of simulated breathing abnormalities is generated using the curve fitting technique for developing a machine learning (ML) classification model. The advantages of generating simulated breathing abnormalities data are two-fold; it will help counter the daunting and time-consuming task of real-time data collection and improve the ML model accuracy. Several ML algorithms are exploited to classify eight breathing abnormalities: eupnea, bradypnea, tachypnea, Biot, sighing, Kussmaul, Cheyne–Stokes, and central sleep apnea (CSA). The performance of ML algorithms is evaluated based on accuracy, prediction speed, and training time for real-time breathing data and simulated breathing data. The results show that the proposed platform for real-time data classifies breathing patterns with a maximum accuracy of 97.5%, whereas by introducing simulated breathing data, the accuracy increases up to 99.3%. This work has a notable medical impact, as the introduced method mitigates the challenge of data collection to build a realistic model of a large dataset during the pandemic.

## 1. Introduction

Coronavirus is a large family of viruses with various forms, such as the Middle East respiratory syndrome coronavirus (MERS-CoV), severe acute respiratory syndrome coronavirus (SARS-CoV), and the latest virus, SARS-CoV-2, also known as COVID-19. COVID-19 infection symptoms include respiratory tract illness, acute viral pneumonia with respirational failure, and death [1]. There are various ways to diagnose COVID-19 infection, of which monitoring breathing rate (BR) is considered one of the most significant indications of COVID-19. Therefore, investigating the BR and its connection with COVID-19 symptoms is now a popular area of research [2]. BR is usually defined as the number of breaths an individual takes per minute when resting. A normal BR is 10 to 24 breaths per minute (bpm), while an abnormal BR for adults can be categorized as hyperventilation (bpm > 24), hypoventilation (bpm < 10), or apnea [3]. For non-COVID scenarios, BR may increase with fever, illness, and other medical conditions.

For COVID scenarios, it is considered very important to determine the BR or breathing activity of the patients as abnormal breathing measurements may indicate a deterioration in the patient’s health [4]. The BR can be measured through manual counting; however, this method is unreliable and prone to error. Therefore, measuring BR usually involves the expertise of a health professional, so it is usually performed in the hospital. The best method for BR measurement in hospitals is spirometry, which calculates the airflow during inhalation and exhalation. Other methods include electrical impedance pneumography (EIP), capnography, and inductance pneumography (IP) [5]. However, these methods require hospitalization. Due to the clinical emergency caused by COVID-19, BR monitoring of the COVID patients increases the risk of virus spread by visiting hospitals. Most patients do not show breathing distress at first, and healthcare professionals must send these patients back home for self-monitoring. According to medical research, patients with minor clinical conditions may worsen in the second week of COVID-19 infection. Therefore, patients with normal breathing functions do not necessitate hospitalizations and must be monitored using telemedicine methods during self-isolation [6]. In contrast, for patients facing acute breathing distress, real-time BR monitoring is very much mandatory.

Breathing can lose its regular rhythm because of numerous medical conditions such as potential injury or metabolic disorders. Breathing abnormalities can have breathing patterns that are shallow, deep, or fast. These abnormal patterns include eupnea, bradypnea, tachypnea, Biot, Kussmaul, sighing, Cheyne–Stokes, and CSA. Eupnea is normal breathing with a uniform pattern and rate, while BR is faster in tachypnea than in eupnea. Deep breaths characterize biot respiration with regular periods of apnea, and Kussmaul is deep and fast breathing. Bradypnea is shallow and slow breathing with a uniform pattern, while sighing is normal breathing punctuated by sighs. Cheyne–Stokes breathing is defined by a gradual increase and decrease in BR, whereas CSA is breathing that repeatedly stops and starts during sleep. Details of the breathing patterns and their causes is given below in Figure 1.

Non-contact monitoring during the pandemic situation is a promising solution to combat the spread of COVID-19. The most significant indication of COVID-19 is abnormal breathing. Therefore, classifying breathing patterns using ML is worthwhile and of great significance. The dataset of breathing patterns required for building the ML model is obtained by assessing test subjects’ breathing patterns. This approach for capturing different breathing patterns yields a limited set of data during pandemic situations, insufficient for developing a reliable ML model. Therefore, there is a need for simulated breathing patterns to overcome the scarcity of real-time breathing data from actual patients. This research collected real-time breathing patterns data through a non-contact approach using the software-defined radio (SDR) platform. Subsequently, a large dataset of simulated breathing patterns was generated using the curve fitting technique. The obtained results were validated by evaluating various ML algorithms based on accuracy, prediction speed, and training time. This approach can be utilized for COVID as well as non-COVID scenarios and has many innovative healthcare applications.

## 2. Literature Review

There are several contact-based and non-contact technologies for breath monitoring presented in the literature. Contact-based technologies require wearable sensors and smartwatches, etc. [7,8]. The devices used in contact-based technologies are expensive, heavy, and are often inconvenient for patients. To avoid this inconvenience, non-contact technologies have also been proposed. The advantages of non-contact technologies include continuous monitoring at home and even during sleep. Most non-contact technologies use camera-based imaging [9] or are RF-based [10]. Camera-based imaging breath monitoring needs a depth camera or thermal imaging camera. There are limitations in camera-based technologies; for example, depth cameras have a high computational cost and are expensive, while thermal imaging is vulnerable to ambient temperature. RF-based non-contact technologies leverage the propagation of electromagnetic (EM) waves that can be extracted through a wireless medium.

Furthermore, RF-based technology for breath monitoring includes various technologies, including radar, Wi-Fi, and SDR. For RF-signal sensing, these technologies can exploit channel state information (CSI) or received signal strength (RSS). There are numerous techniques for radar-based breath monitoring, including the Doppler radar [11] and frequency-modulated continuous-wave (FMCW) [12]. These radar-based techniques require high-cost, specialized hardware working at high frequency. Furthermore, the Vital-Radio system [13] uses a FMCW radar to track breathing and heart rates with a wide bandwidth from 5.46 GHz to 7.25 GHz. For Wi-Fi-based breath monitoring, numerous approaches using RSS and CSI are mentioned in the literature. The authors of [14] explored the use of RSS measurements on the links between wireless devices to find the BR and location of a person in a home environment. Similarly, a complete architecture for finding breathing signals from noisy Wi-Fi signals was presented in [15]. A further study [16] provided a non-contact CSI-based breath monitoring system. Schmidt [17] applied the Hampel filter on the CSI series to eliminate outliers and high-frequency noises, and then, BR was measured by performing FFT on all the CSI streams targeting frequencies between 0.1 Hz and 0.6 Hz. Wang et al. [18] proposed a BR monitoring system using the CSI through a single pair of commercial Wi-Fi devices. Wi-Fi-based RF sensing has several advantages, such as cost-effectiveness and ready availability. However, it also has disadvantages, such as lack of scalability and flexibility, and under-reporting Orthogonal Frequency Division Multiplexing (OFDM) subcarriers [19]. 

SDR-based breath monitoring has been investigated by various authors [20,21,22,23,24]. SDR-based breath monitoring is considered the most efficient among all the RF-based techniques, as it offers a flexible, portable, and scalable solution. Additionally, this technology permits the selection of the operating frequency and transmitted/received power. Moreover, it allows the simple execution of signal processing algorithms. ML has also been exploited for breath monitoring to help accurately classify various breathing abnormalities. Several authors have used ML for classifying breathing patterns [25]. However, several studies were unable to obtain reasonable accuracy, and some were only successful in classifying basic breathing patterns, including normal and fast breathing [26]. Consequently, there is a need for a platform to monitor and accurately classify a diverse range of breathing patterns. The summary of various technologies discussed in the literature is shown in Figure 2.

## 3. System Architecture

The system architecture consists of four layers, as shown in Figure 3. The functionality of each layer is explained below:

### 3.1. Data Extraction Layer

The first layer is the data extraction layer, which is responsible for breathing data extraction. There are three main blocks in this layer containing the transmitter, receiver, and wireless channel. The transmitter contains transmitter PC and transmitter universal software radio peripheral (USRP). First, random data bits are generated and mapped to quadrature amplitude modulation (QAM) symbols in the transmitter PC. Then, these QAM symbols are further split into parallel frames. After that, reference data symbols are inserted in each parallel frame. On the receiver side, the reference symbols will be used for channel estimation. Then in each frame, zeros are positioned at the edges and 1 zero at DC. Next, frequency domain signals are converted into time-domain signals by applying inverse fast Fourier transform (IFFT). Subsequently, a cyclic prefix (CP) is introduced in every frame by repeating the last one-fourth of the points at the start. On the receiver side, the CP will be used in the elimination of frequency and time offset.

Then, the host PC sends this data to the USRP kit through gigabit ethernet. First, this data is digitally upconverted and translated into an analog form using a digital upconverter (DUC) and digital to analog converter (DAC), respectively. The USRP then passes this analog signal through a low pass filter (LPF) and mixes it up to a user-specified frequency. Before transmitting the resultant signal using an omnidirectional antenna, USRP moves it through a transmit amplifier for gain adjustment. After passing through the wireless channel, the transmitted signal is received at the receiver USRP device using an omnidirectional antenna. This received signal is then moved through a low noise amplifier (LNA) and drive amplifier (DA) for noise element elimination and gain adjustment, respectively. The subsequent signal is passed through LPF, then an analog to digital converter (DAC), followed by the digital down converter (DDC) for filtering and decimating the signal. Eventually, the resultant signal is moved to the host PC using a gigabit ethernet cable. Now CP is eliminated from each frame at the host PC; additionally, time and frequency offset are removed by applying the Van de Beek algorithm [27]. Afterward, the fast Fourier transform (FFT) converts the time domain samples into frequency domain symbols. Finally, breathing patterns are detected by extracting the amplitude response of the frequency domain signal.

The wireless channel for the OFDM system can be regarded as a narrowband flat fading channel, which can be represented in the frequency domain, as shown in Equation (1):(1)$$\bar{Y}=H\times \bar{X}+\bar{N}$$
where $\bar{Y}$ and $\bar{X}$ denote the received and transmitted wireless signal vectors, respectively, $\bar{N}$ is the additive white Gaussian noise, and H represents the OFDM channel frequency response for all subcarriers, and this H can be estimated from $\bar{Y}$ and $\bar{X}$. Here, the OFDM system uses 256 subcarriers for data transmission on a 20 MHz channel. The channel frequency response for all subcarriers can be represented by Equation (2) as:(2)$$H=\left[\begin{array}{cccc} H_{11} & H_{12} & \ldots & H_{1s}\\ H_{21} & H_{22} & \ldots & H_{2s}\\ \vdots & \vdots & \ldots & \vdots \\ H_{k1} & H_{k2} & \ldots & H_{ks} \end{array}\right]$$
where k represents the OFDM subcarriers, and s represents acquired samples. The frequency response of the channel for a single subcarrier i is denoted by $H_{i}$, and is a complex value, given in Equation (3) as:(3)$$H_{i}=\left|H_{i}\right|\;\exp\left(j∠H_{i}\right)$$
where $\left|H_{i}\right|$ and $∠H_{i}$ are the amplitude and phase response of OFDM subcarrier i, respectively. For indoor lab environments with multipath components, the channel frequency response $H_{i}$ of subcarrier i is expressed in Equation (4) as:(4)$$H_{i}=\overset{N}{\underset{n=0}{\sum }}r_{n}\cdot e^{-j2\pi f_{i}\tau_{n}}$$
where N is the total number of multipath components, and $r_{n}$ and $\tau_{n}$ are the attenuation and propagation delay on the $n_{th}$ path, respectively, while $f_{i}$ represents the frequency of the $i_{th}$ subcarrier.

### 3.2. Data Preprocessing Layer

Raw CSI data received from the extraction layer is sent to the data preprocessing layer. This layer is further divided into four sublayers.

#### 3.2.1. Subcarrier Selection

The first step in the data preprocessing layer is subcarrier selection. After each activity, 256 OFDM subcarriers are acquired at the receiver. It is realized that the susceptibility of each subcarrier is distinct for the breathing experiment. Therefore, for good detection, eliminating all subcarriers that are less susceptible to breathing activity is necessary. Therefore, the variance of all subcarriers is measured. Based on this, all subcarriers that are less susceptible to breathing activity are eliminated, as shown in Figure 4a for all OFDM subcarriers.

#### 3.2.2. Outliers Removal

After subcarrier selection, wavelet filtering is applied to eliminate the outliers from raw data by retaining sharp transition. This can be seen in Figure 4b for all OFDM subcarriers. For wavelet filtering, “scaled noise option” and “soft heuristic SURE” thresholding is used on coefficients by choosing “syms5” and “level 4”.

#### 3.2.3. Data Smoothing

The moving average filter of “size 8” is applied for data smoothing, which removes high-level frequency noise, and this can be seen in Figure 4c for all OFDM subcarriers. The output of the moving average filter of window “size 8” can be represented by Equation (5):(5)$$y\left[n\right]=\frac{1}{N}\overset{N-1}{\underset{i=0}{\sum }}x\left[n-i\right]$$
where y[n] is the current output, x[n] is the current input, and N is the window size of the moving average filter.

#### 3.2.4. Data Normalization

Finally, waveform data is normalized to the maximum and minimum value to 1 and −1, respectively, by using the following Equation (6):(6)$$\bar{y\left[n\right]}=\frac{y\left[n\right]-offset}{scale}$$
where $\bar{y\left[n\right]}$ is the normalized data, and y[n] is the input data. Here, input waveform data is scaled and offset by some values to acquire normalized waveform. For example, the normalized waveform for a single OFDM subcarrier is shown in Figure 4d. After performing the above steps, processed CSI data is obtained to classify breathing patterns through ML algorithms.

### 3.3. Data Simulation Layer

The amount of real-time breathing experimentation is not sufficient to train a robust ML classification model. Therefore, a simulation model inspired by [28] was developed to overcome the data scarcity issue. Based on the characteristics of actual real-time breathing patterns data, a simulation model was developed to generate abundant and high-quality simulated breathing data. As breathing is a continuous process of inhalation and exhalation, breathing signals measured by the non-contact method can be approximated by the sinusoidal waveforms. Here, breathing patterns are simulated through the curve fitting function available in MATLAB. The curve fitting is usually performed to theoretically describe experimental data points with a model (equation or function) and acquire the model’s parameters. In this research work, the curve fitting function of MATLAB is used to model and generate all breathing patterns [29]. As all breathing patterns can be represented by sinusoidal waveform, therefore by using the curve fitting function of MATLAB, real-time breathing patterns are modelled by the sum of seven sinusoidal terms, which can be represented by Equation (7):(7)$$y=\overset{n}{\underset{i=0}{\sum }}a_{i}\sin(b_{i}x+c_{i})$$
where a is the amplitude, b is the frequency, and c is the phase for each sinusoidal term, while x represents OFDM samples, and n is the total number of sinusoidal terms in the summation. The coefficients’ values for eight breathing patterns are shown in Table 1. Here, for illustration purposes, simulated and real-time Biot breathing patterns are shown in Figure 5. Real-time breathing patterns are prone to fluctuations; therefore, to make simulated breathing patterns closer to real-time data, the additive white Gaussian noise (AWGN) function available in MATLAB introduces noise into the simulated breathing patterns [30]. A huge amount of simulated data can be generated by slight variations in AWGN values or the coefficients’ values shown in Table 1. At the output of the data simulation layer, simulated CSI for eight breathing patterns is generated and used for classification purposes.

### 3.4. Data Classification Layer

In this layer, the processed and simulated CSI is used for training and testing purposes. First various statistical features of the processed CSI are extracted, and the performance of the ML algorithms is evaluated based on accuracy, prediction speed, and training time. Likewise, statistical features for the simulated CSI are extracted. After this, the performance of the ML algorithms is evaluated for the simulated data. The details of the statistical features are shown in Table 2. As the accuracy of ML algorithms depends upon the size and type of the dataset, it can be enhanced by enlarging the dataset. In this work, the dataset size was enhanced by introducing a large amount of simulated breathing data. Furthermore, random five-fold cross-validation was applied for classification purposes.

## 4. Results and Discussion

In this section, the experimental setup is presented, and the results are discussed.

### 4.1. Experimental Setup

The experimental setup monitors and classifies eight breathing patterns by detecting small-scale activities in a real-time wireless medium through acquiring fine-grained CSI. The experimental setup contains three main blocks: the transmitter, receiver, and wireless channel, as shown in Figure 6. The transmitter block contains a transmitter PC, and the USRP model 2922 is utilized as SDR hardware to perform generic RF functionality. In contrast, the receiver block consists of a receiver PC and receiver USRP. Omnidirectional antennas are also used to capture the variation in the CSI due to breathing. The RF signal generated by the transmitter reaches the receiver through multipaths in the indoor lab environment. When an individual is present in the lab environment, an additional path is created due to the human body’s diffraction or reflection of signals. Therefore, the impact of human breathing on the signal’s propagation is acquired on the receiver side in the form of CSI.

Five volunteers participated in the study and performed the breathing patterns; their details are given in Table 3. Each volunteer was sitting in a relaxed position. Both the transmitter and receiver USRPs were positioned parallel to the abdomen of the volunteer at a 1-m distance. All volunteers were professionally trained to perform each breathing pattern. Ten datasets were collected from five volunteers for eight breathing patterns to perform 400 experiments. Each breathing activity was performed for 30 s by each volunteer.

### 4.2. Breathing Patterns’ Monitoring

This section describes the monitoring of breathing patterns using SDR-based RF sensing. The CSI amplitude response was exploited to analyze the breathing patterns. The variations in amplitude frequency response were observed for each breathing experiment over 3500 OFDM samples. For illustration purposes, the results from subject 2 for eight breathing patterns are depicted in Figure 7 for a single OFDM subcarrier. Eupnea is normal breathing with a uniform rate and pattern with a BR between 12 to 24 bpm. From Figure 7a, it can be observed that there are 12 breaths in a half-minute, which lies within the range of normal breathing. Bradypnea is shallow and slow breathing with a uniform pattern, as can be observed in Figure 7b, showing 6 breaths in a half-minute, which lies in the range of slow breathing. In tachypnea, the BR is faster than eupnea, which can be verified in Figure 7c, which shows there are 15 breaths in a half minute. Biot is characterized by deep breathing with regular periods of apnea, and Figure 7d depicts deep breaths followed by apnea. Sighing is normal breathing punctuated by sighs. This can be observed in Figure 7e, which shows normal breathing punctuated by frequent deep breaths. Cheyne–Stokes is defined by a gradual increase and decrease in BR, and Figure 7f clearly shows a gradual increase and decrease in BR. Kussmaul is deep and fast breathing, which can be observed in Figure 7g. Furthermore, CSA is a type of breathing in which breathing repeatedly stops and starts during sleep and this can be seen in Figure 7h, which shows periods of no breathing during normal breathing.

### 4.3. Breathing Patterns’ Classification

In this section, the results of ML algorithms for the classification of breathing patterns are discussed. To assess the performance of each ML algorithm, a confusion matrix was used, with eight predicted and true classes. The diagonal entries of the matrix represent the cases where the actual class and predicted class are matched. The cell values other than the diagonal entries show where the ML algorithm performed poorly. The performance of the ML algorithms was evaluated based on prediction speed, accuracy, and training time. Prediction speed is measured as observations per second, accuracy is calculated as a percentage, and training time is measured in seconds. Initially, the real-time breathing patterns data were trained by four ML algorithms. The confusion matrix results are shown in Table 4, and it can be seen that all algorithms classified these breathing patterns successfully. Then, data from ten thousand simulated breathing patterns were trained through the ML algorithm. Finally, the confusion matrix results are shown in Table 5, and it can be seen that these algorithms can classify breathing patterns even more successfully.

Finally, in Table 6, a performance comparison is shown for real-time and simulated breathing data. It can be observed that accuracy is improved for all algorithms for simulated breathing data compared to real-time breathing data. For example, for cosine nearest neighbor classifiers (KNN), accuracy is increased from 97.5% to 99.3%, while for complex tree algorithms, accuracy is increased from 96.8% to 98.4%, and for ensemble boosted tree algorithms, the accuracy is increased from 85.6% to 94.7%. Furthermore, for linear support vector machine (SVM) algorithms, the accuracy is increased from 75.5% to 84.9%. The performance comparison of ML algorithms in terms of accuracy is also elaborated in Figure 8. It can be observed that accuracy is improved for all algorithms when the breathing patterns dataset is increased through simulation.

## 5. Conclusions

In this article, a solution is proposed for the problem of collecting a large dataset of abnormal breathing patterns during the pandemic. This approach improves the classification accuracy of the ML model. A non-contact SDR platform was used to collect real-time breathing patterns data based on variations in CSI. The real-time data was used to generate a large, simulated dataset through the curve fitting technique. This increases the dataset size, which will help in building a reliable ML model. Different ML algorithms were exploited for breathing patterns classification for real-time as well as simulated breathing data. It was verified that accuracy improves by introducing simulated data for training purposes. This research work can be used for COVID and non-COVID scenarios. The results indicate that the developed platform is accurate and robust for monitoring human breathing, and its accuracy can be further enhanced by introducing more simulated breathing data. The future applications of this research work are numerous, for example, it can be used as a pre-examination tool for patients to provide clues about the nature of the illness, it can be used in homes for individual monitoring during the day, and it can even be deployed in public places for infection monitoring in crowds. This work has a few limitations, including that experiments were performed on single subjects in an indoor lab environment, and actual patients were not chosen for data collection. So, future recommendations for this research work would remove all of the limitations stated in this research.

## Figures and Tables

**Figure 1 sensors-21-06750-f001:**
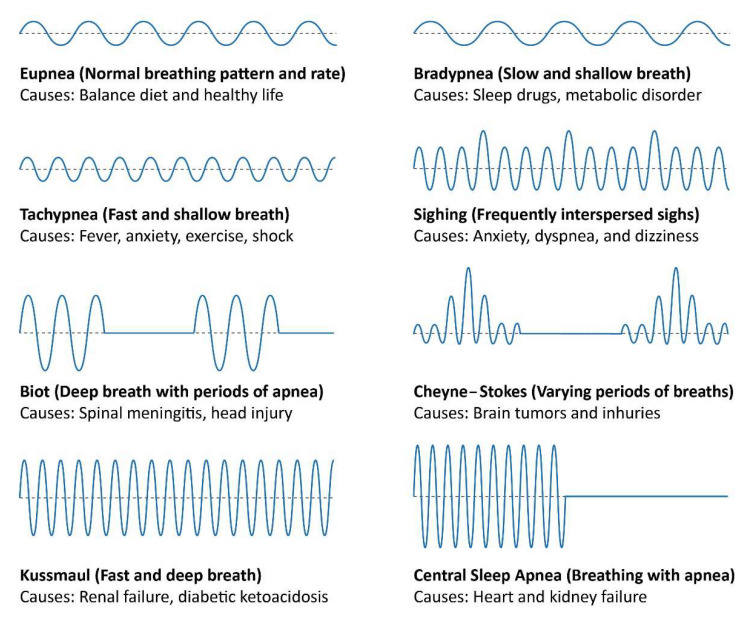
Breathing patterns and causes.

**Figure 2 sensors-21-06750-f002:**
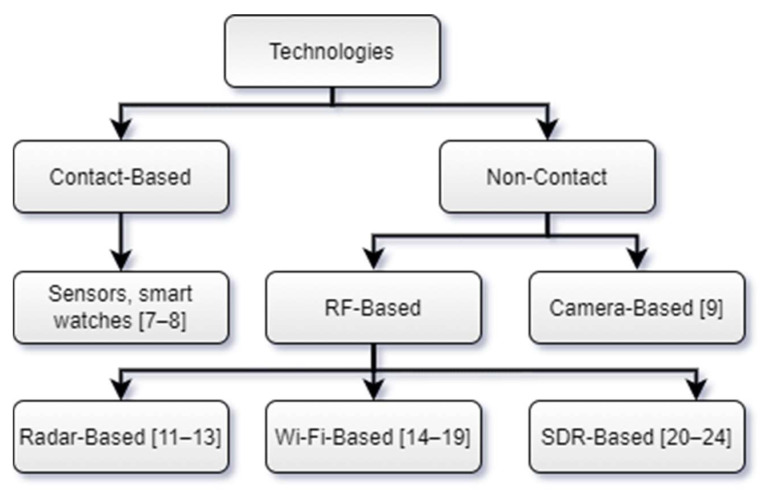
Literature review summary.

**Figure 3 sensors-21-06750-f003:**
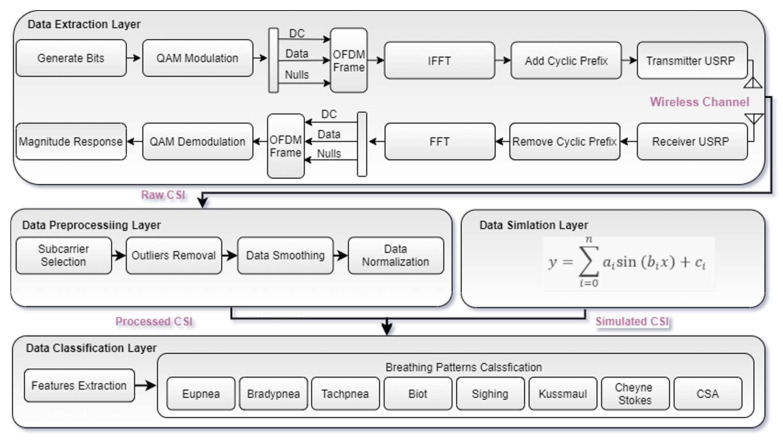
System architecture.

**Figure 4 sensors-21-06750-f004:**
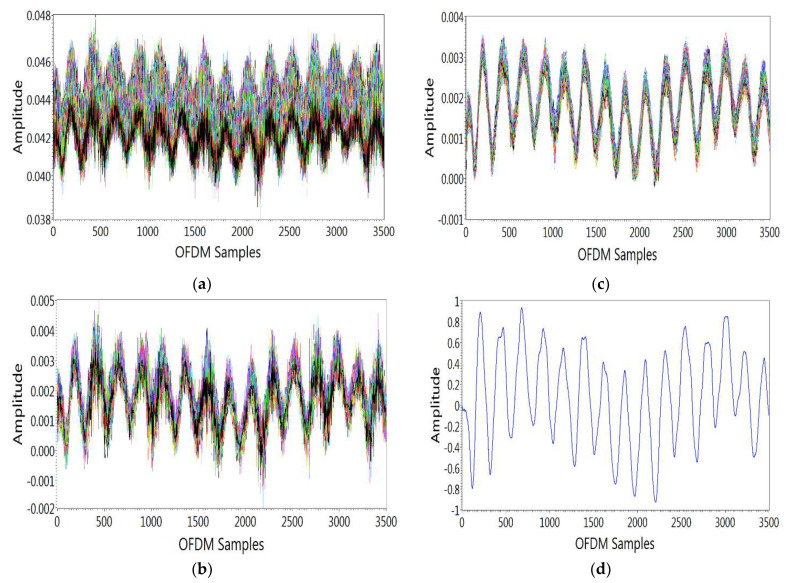
Data preprocessing layer. (**a**) Subcarrier selection, (**b**) Outliers removal, (**c**) data smoothing, (**d**) data normalization.

**Figure 5 sensors-21-06750-f005:**
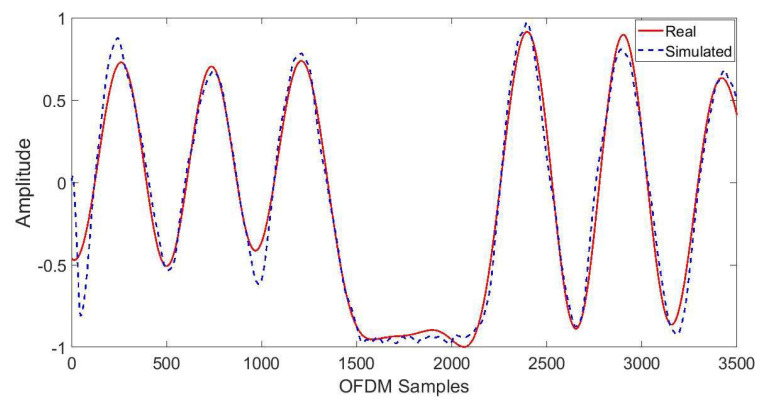
Simulated and real-time Biot breathing.

**Figure 6 sensors-21-06750-f006:**
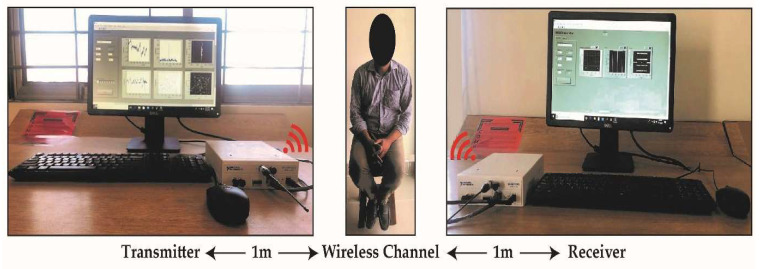
Non-contact breathing sensing experimental setup.

**Figure 7 sensors-21-06750-f007:**
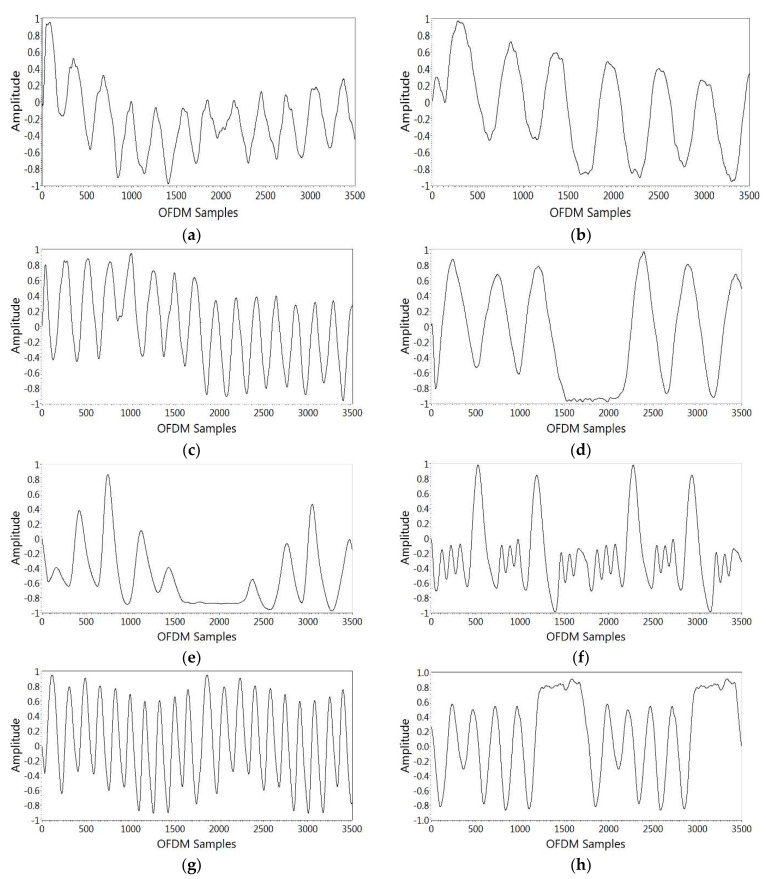
Breathing pattern results. (**a**) Eupnea, (**b**) bradypnea, (**c**) tachypnea, (**d**) Biot, (**e**) sighing, (**f**) Cheyne–Stokes, (**g**) Kussmaul, (**h**) CSA.

**Figure 8 sensors-21-06750-f008:**
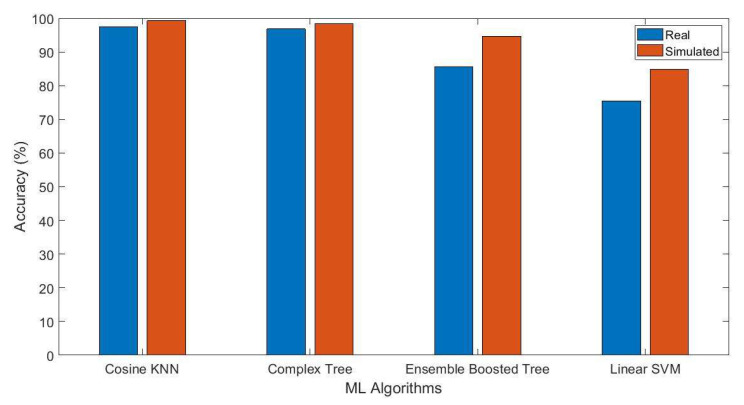
Comparison of the algorithms’ accuracy.

**Table 1 sensors-21-06750-t001:** Coefficients values for simulated breathing patterns.

Coefficients Values		Breathing Patterns							
		Eupnea	Bradypnea	Tachypnea	Biot	Sighing	Kussmaul	Cheyne–Stokes	CSA
Amplitude	a1	0.506	0.409	0.406	0.435	0.256	0.547	0.775	0.427
	a2	0.342	0.303	0.326	0.260	0.333	0.282	0.302	0.437
	a3	0.064	0.441	3.852	0.179	0.296	0.198	0.232	0.176
	a4	0.077	0.174	1.838	0.564	0.192	0.192	6.850	0.329
	a5	0.218	0.109	3.693	0.458	0.147	0.123	6.855	0.240
	a6	0.043	0.075	0.066	0.237	0.179	0.120	0.205	0.202
	a7	0.066	0.607	1.692	0.789	0.152	9.275	0.115	0.164
Frequency	b1	0.001	0.011	0.027	0.002	0.001	0.036	0.001	0.004
	b2	0.021	0.012	0.002	0.015	0.018	0.039	0.016	0.025
	b3	0.005	0.001	0.025	0.010	0.011	0.003	0.004	0.000
	b4	0.019	0.008	0.029	0.012	0.021	0.032	0.021	0.007
	b5	0.003	0.014	0.025	0.004	0.004	0.025	0.021	0.018
	b6	0.029	0.005	0.009	0.007	0.029	0.029	0.018	0.022
	b7	0.022	0.001	0.029	0.001	0.007	0.000	0.005	0.011
Phase	c1	2.388	−1.035	−0.297	3.864	−2.571	−2.797	2.968	2.462
	c2	−0.373	2.775	0.207	−2.694	−1.710	−0.073	1.868	2.009
	c3	−1.513	3.456	2.941	3.302	2.123	0.331	−2.013	0.944
	c4	1.107	−1.130	0.915	−0.770	2.059	−2.012	−3.590	−2.466
	c5	2.434	1.926	−0.119	−2.595	−1.478	−1.583	−0.497	2.735
	c6	−1.227	2.053	−0.194	−0.835	−1.264	−1.548	1.154	−1.577
	c7	−0.075	1.091	4.008	−4.899	−1.650	3.133	−1.390	−1.952

**Table 2 sensors-21-06750-t002:** Statistical features.

Sr. No.	Statistical Features	Detail	Equations
1	Minimum	Minimum value in data	$X_{min}=min\left(x_{k}\right)$
2	Maximum	Maximum value in data	$X_{max}=max\left(x_{k}\right)$
3	Mean	Data mean	$X_{m}=L\overset{L}{\underset{k=1}{\sum }}x_{k}$
4	Variance	Spread of data	$X_{S\;D}=\overset{n}{\underset{k=1}{\sum }}{\left(x_{k}-x_{m}\right)}^{2}$
5	Standard deviation	Square root of variance	$X_{v}=\sqrt[{2}]{\frac{1}{L-1}\overset{L}{\underset{k=1}{\sum }}{\left(x_{k}-X_{m}\right)}^{2}}$
6	Peak-to-peak value	Variations in data about the mean	$X_{p-p}=X_{max}-X_{min}\left(k=1,2,\ldots ,L\right)$
7	RMS	Root mean of square data	$X_{RMS}=\sqrt[{2}]{\frac{1}{L}\overset{L}{\underset{k=1}{\sum }}x_{k}{}^{2}}$
8	Kurtosis	Peak sharpness of a frequency–distribution curve	$X_{K}=\frac{\frac{1}{L}\sum_{k=1}^{L}{\left(\left|\begin{array}{c} x_{k} \end{array}\right|-X_{m}\right)}^{4}}{X_{RMS}{}^{4}}$
9	Skewness	Measure of symmetry in data	$X_{S}=\frac{\frac{1}{L}\sum_{k=1}^{L}{\left(\left|\begin{array}{c} x_{k} \end{array}\right|-X_{m}\right)}^{3}}{X_{RMS}{}^{3}}$
10	Interquartile range	Mid-spread of data	$X_{IQ}=X_{3}-X_{1}$
11	Waveform factor	Ratio of the RMS value to the mean value	$X_{W}=\frac{X_{RMS}}{X_{M}}$
12	Peak factor	Ratio of maximum value of data to RMS	$X_{P}=\frac{max\left(x_{k}\right)}{X_{RMS}}\;\left(k=1,2,\ldots ,L\right)$
13	FFT	Frequency information about data	$X_{FFT}=\overset{L}{\underset{k=-L}{\sum }}x\left(n\right)e^{-j\frac{2\pi }{N}nk}$
14	Frequency Min	Minimum frequency component	$X_{fmin}=Min\left(X_{FFT}\right)$
15	Frequency Max	Maximum frequency component	$X_{fmax}=Max\left(X_{FFT}\right)$
16	Spectral Probability	Probability distribution of spectrum	$X_{SP}=\frac{FFT{\left(d\right)}^{2}}{\sum_{k=-L}^{L}FFT{\left(k\right)}^{2}}$
17	Spectrum Entropy	Measure of data irregularity	$X_{H}=\overset{L}{\underset{k=-L}{\sum }}p\left(d\right)\ln\left(p\left(d\right)\right)$
18	Signal Energy	Measure of energy component	$X_{SE}=\overset{L}{\underset{k=-L}{\sum }}{\left|p\left(d\right)\right|}^{2}$

**Table 3 sensors-21-06750-t003:** Details of volunteers.

Sr. No.	Gender	Age (Years)	Weight (Pounds)	Height (Inches)	Body Mass Index
1	Male	26	168	68	25.4
2	Male	28	144	71	20.3
3	Male	31	114	70	16.8
4	Male	31	113	69	16.6
5	Male	31	143	68	21.5

**Table 4 sensors-21-06750-t004:** Confusion matrix for real-time breathing data.

Algorithms	Actual/Predicted	Eupnea	Bradypnea	Tachypnea	Biot	Sighing	Kussmaul	Cheyne–Stokes	CSA
Cosine KNN	Eupnea	3487	65	63	9	0	4	7	15
	Bradypnea	58	3512	42	7	0	10	0	21
	Tachypnea	134	71	3396	24	0	1	13	11
	Biot	3	2	12	3620	3	4	6	0
	Sighing	0	0	0	0	3648	2	0	0
	Kussmaul	2	6	1	17	3	3618	3	0
	Cheyne–Stokes	0	0	0	8	0	4	3638	0
	CSA	43	29	31	1	0	0	0	3546
Complex Tree	Eupnea	3491	74	0	76	0	0	9	0
	Bradypnea	48	3596	0	4	0	0	2	0
	Tachypnea	0	0	3487	0	0	22	0	141
	Biot	54	3	0	3588	0	0	5	0
	Sighing	0	0	0	0	3649	1	0	0
	Kussmaul	0	0	55	0	2	3587	0	6
	Cheyne–Stokes	8	2	0	19	0	0	3621	0
	CSA	0	0	407	0	0	0	0	3243
Ensemble Boosted Tree	Eupnea	2122	613	0	781	0	0	134	0
	Bradypnea	101	3007	0	62	0	0	480	0
	Tachypnea	0	0	3372	0	0	162	0	116
	Biot	99	42	0	3478	0	0	31	0
	Sighing	0	0	0	0	3644	6	0	0
	Kussmaul	0	0	25	0	149	3461	0	15
	Cheyne–Stokes	6	0	0	282	0	0	3362	0
	CSA	0	0	1073	0	0	36	0	2541
Linear SVM	Eupnea	2367	631	138	272		56	97	89
	Bradypnea	712	1958	174	476	118	93	43	76
	Tachypnea	2	0	2967	14	0	123	0	544
	Biot	519	497	32	2264	0	267	70	1
	Sighing	0	0	0	0	3449	201	0	0
	Kussmaul	0	0	50	191	71	3314	3	21
	Cheyne–Stokes	192	143	0	121	136	280	2778	0
	CSA	0	0	696	0	0	0	0	2954

**Table 5 sensors-21-06750-t005:** Confusion matrix for simulated breathing data.

Algorithms	Actual /Predicted	Eupnea	Bradypnea	Tachypnea	Biot	Sighing	Kussmaul	Cheyne–Stokes	CSA
Cosine KNN	Eupnea	13,493	43	77	3	0	7	4	23
	Bradypnea	33	13,539	42	2	0	22	0	12
	Tachypnea	179	72	13,334	49	0	1	7	8
	Biot	4	3	36	13,604	0	3	0	0
	Sighing	0	0	0	0	13,648	0	2	0
	Kussmaul	3	5	0	1		1340	1	0
	Cheyne–Stokes	2	0	0	3	1	1	13,642	1
	CSA	49	29	13	21	0	8	6	13,523
Complex Tree	Eupnea	12,960	318	0	365	0	0	7	0
	Bradypnea	68	13,491	0	84	0	0	2	5
	Tachypnea	1	0	13,352	0	0	94	0	203
	Biot	46	25	0	13,570		1	8	
	Sighing	0	0	0	0	13,629	21	0	0
	Kussmaul	0	0	14	0	1	13,621	0	3
	Cheyne–Stokes	113	12	0	45	0	0	13,480	0
	CSA	0	0	358	0	2	3	0	13,287
Ensemble Boosted Tree	Eupnea	12,580	798	0	130	0	0	142	0
	Bradypnea	375	12,955	0	0	0	0	320	0
	Tachypnea	0	0	13,252	0	0	325	0	73
	Biot	1003	496	0	12,134	0	0	18	0
	Sighing	0	0	0	0	13,499	151	0	0
	Kussmaul	0	0	1	0	152	13,456	0	41
	Cheyne–Stokes	127	61	0	6	0	0	13,456	0
	CSA	0	0	1377	0	120	59	0	12,094
Linear SVM	Eupnea	10,003	0	0	0	3647	0	0	0
	Bradypnea	1	9998	0	2	3649	0	0	0
	Tachypnea	17	0	10,027	0	3606	0	0	0
	Biot	0	0	3	10,000	3621	0	26	0
	Sighing	149	0	0	0	13,496	0	5	0
	Kussmaul	0	0	0	0	3650	10,284	0	0
	Cheyne–Stokes	0	0	0	0	3650	0	10,000	0
	CSA	0	0	0	0	3650	0	0	10,000

**Table 6 sensors-21-06750-t006:** Performance of ML algorithms.

Algorithms	Real-Time Breathing Data			Simulated Breathing Data		
	Accuracy (%)	Prediction Speed (obs/s)	Training Time (s)	Accuracy (%)	Prediction Speed (obs/s)	Training Time (s)
Cosine KNN	97.5	~2200	306.35	99.3	~500	2583.60
Complex Tree	96.8	~410,000	11.16	98.4	~86,000	140.77
Ensemble Boosted Tree	85.6	~80,000	390.58	94.7	~44,000	2897.90
Linear SVM	75.5	~98,000	219.75	84.9	~32,000	1184.90

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
