# Peer review of "Improving Machine Learning Classification Accuracy for Breathing Abnormalities by Enhancing Dataset"

_sensors, 2021, doi:10.3390/s21206750_

Round 1

Reviewer 1 Report

The research focus on monitoring breathing patterns so that distinguish the abnormalities in the. The breathing patterns were measured using RF sensors. The signal from sensors were then analysed using machine learning algorithms to classify the beathing abnormalities in one of eight breathing abnormalities. The research focused mainly on developing algorithms to classify the breathing abnormalities while RF sensor just was mentioned with a little information. The authors should address the following in order to be published

  1. Literature review section should be removed and replaced by presenting in detail the sensors used in this research. Doe the sensors affect the algorithms using in this research?
  2. There should be comparisons of algorithms using in this research and in literatures. What are the advantages and novelties of the algorithms in this research compared with those in literatures? How is the performance of the algorithms in this research compared with those in literatures?
  3. How do the algorithms work to classify the breathing abnormalities when patients’ respiration frequently changes?
  4. How can the algorithms eliminate noises from measurement signals?
  5. Is performance of the algorithms better over time (learning from previous measurement)? and how

Reviewer 2 Report

sensors and smart watches classified as invasive is not inline with the general definition of the term. 

Why is random data bits inserted  into the channel and not a known sequence ?

How many subcarriers,  what frequency range and bandwidth,  how big a cycling prefix ?

When the data is aquired,  is there a single person or multiple people in the area ?

You have created an oFDM symbol of unknown subcarriers with an unknown number of pilots and unspecied length of CP.  Yet you claim that the amplitude response of the received signal is used to measure BR.  How does this happen, why does breathing effect amplitude respnse ?

It is not clear what makes some subcarriers susceptible oto breating and not others. 

Figure 4 is unclear. do not understand what does OFDM samples mean, are they the plot of teh same subcarrier over time? In the same fiugure outlier removal , data smothening are not clear

Section 3.3 should be where the main focus of this reserarch is base don the title.  I am not sure how the work in this section is based on the measurements made in the earlier section

Equation 7  needs revision.  c sub i should be in the pranthesis of the sign term. It is the phase I assume

How did you collect data for each one of the cases specified in table 1,  if all simulated how did you base the simulated data generation  on experimentally collected data. 

section3.4 appears twice in the paper 

Table 2,  lists well known formulas. 

Table 3 has repetitions ,  weight(Pounds) is repeated 

Table 3 only lists 5 subjects.  Two small a number to make any generalization. 

Reviewer 3 Report

The article "Improving Machine Learning Classification Accuracy for Breathing Abnormalities by Enhancing Dataset" presents RF-based technology is used for collecting breathing abnormalities data and subsequently, based on this data is identified breathing abnormalities using a machine learning (ML) classification model. In this reviewer’s opinion, the article needs improvement:

1- The concept of using RF technology with Machine Learning to identify COVID-19 symptoms like coughing is already used in article 10.1109/JSEN.2020.3028494. However, this current article proposes important innovations. Please insert a reference to article 10.1109/JSEN.2020.3028494, 10.3390/s21093172.

2- It is important to clarify the advantages and disadvantages of using the technique with 2 SDR devices (as proposed in the article) instead of using radar equipment with a single device.

3- Insert photos of the experimental setup.

4- Why use an omnidirectional antenna instead of a directional antenna if exist the region of interest (one person)?

Round 2

Reviewer 1 Report

The revised manuscript is quite satisfactory from my perspective.

Reviewer 3 Report

Nothing to add.